# Attention Deficit-Hyperactivity Disorder (ADHD): From Abnormal Behavior to Impairment in Synaptic Plasticity

**DOI:** 10.3390/biology12091241

**Published:** 2023-09-15

**Authors:** Gonzalo Ugarte, Ricardo Piña, Darwin Contreras, Felipe Godoy, David Rubio, Carlos Rozas, Marc Zeise, Rodrigo Vidal, Jorge Escobar, Bernardo Morales

**Affiliations:** 1Laboratory of Neuroscience, Department of Biology, Faculty of Chemistry and Biology, University of Santiago of Chile, Santiago 9170022, Chile; gonzalo.ugarte@usach.cl (G.U.); darwin.contrerasp@usach.cl (D.C.); felipe.godoy.g@usach.cl (F.G.); david.rubio@usach.cl (D.R.); carlos.rozas@usach.cl (C.R.); 2Department of Biology, Faculty of Sciences, Metropolitan University of Education Sciences, Santiago 7760197, Chile; ricardo.pina@umce.cl; 3Department of Human Sciences, Faculty of Human Science, Bernardo O’Higgins University, Santiago 8370854, Chile; 4School of Psychology, Faculty of Humanities, University of Santiago of Chile, Santiago 9170022, Chile; marc.zeise@usach.cl; 5Laboratory of Genomics, Molecular Ecology and Evolutionary Studies, Department of Biology, Faculty of Chemistry and Biology, University of Santiago of Chile, Santiago 9170022, Chile; ruben.vidal@usach.cl; 6Instituto de Química, Pontificia Universidad Católica de Valparaiso, Valparaíso 2340000, Chile

**Keywords:** neurodevelopmental disorders, attention deficit-hyperactivity disorder, hippocampus, long-term potentiation, dendritic spines, prenatal nicotine exposure model

## Abstract

**Simple Summary:**

Attention deficit-hyperactivity disorder (ADHD) is a neurodevelopmental disorder with high incidence in children and adolescents characterized by hyperactivity, impulsivity, and inattention. Neuroanatomical anomalies such as the volume reduction in the neocortex and hippocampus and the abnormal dendritic spine pruning during postnatal development are shared by several neuropsychiatric diseases such as schizophrenia, autism spectrum disorder and ADHD. This review presents recent evidence focused on the molecular mechanisms involved in dendritic spine remodeling in the context of synaptic plasticity and learning. The impairment in synaptic plasticity, working memory and spine phenotype in a murine model of ADHD are also discussed.

**Abstract:**

Attention deficit-hyperactivity disorder (ADHD) is a neurodevelopmental disorder with high incidence in children and adolescents characterized by motor hyperactivity, impulsivity, and inattention. Magnetic resonance imaging (MRI) has revealed that neuroanatomical abnormalities such as the volume reduction in the neocortex and hippocampus are shared by several neuropsychiatric diseases such as schizophrenia, autism spectrum disorder and ADHD. Furthermore, the abnormal development and postnatal pruning of dendritic spines of neocortical neurons in schizophrenia, autism spectrum disorder and intellectual disability are well documented. Dendritic spines are dynamic structures exhibiting Hebbian and homeostatic plasticity that triggers intracellular cascades involving glutamate receptors, calcium influx and remodeling of the F-actin network. The long-term potentiation (LTP)-induced insertion of postsynaptic glutamate receptors is associated with the enlargement of spine heads and long-term depression (LTD) with spine shrinkage. Using a murine model of ADHD, a delay in dendritic spines’ maturation in CA1 hippocampal neurons correlated with impaired working memory and hippocampal LTP has recently reported. The aim of this review is to summarize recent evidence that has emerged from studies focused on the neuroanatomical and genetic features found in ADHD patients as well as reports from animal models describing the molecular structure and remodeling of dendritic spines.

## 1. Introduction

Attention deficit-hyperactivity disorder (ADHD) is a neurodevelopmental disorder with a high incidence in children associated with decreased levels of dopamine and norepinephrine in the brain. The current pharmacotherapy uses mainly inhibitors of dopamine and norepinephrine transporters such as the psychostimulants methylphenidate and amphetamine and the non-stimulant atomoxetine. ADHD is considered a neurodevelopmental disorder in a similar manner to autism spectrum disorder (ASD) and intellectual disability (ID). Early studies of postmortem brain samples of patients with ASD and ID showed an abnormal density of dendritic spines in cortical neurons associated with a putative defective or enhanced spine pruning during postnatal development [1,2].

It is well known that the molecular machinery involved in synaptic transmission is enriched in the dendritic spines, that appear as small protuberances emerging from the dendritic shaft. These structures are remodeled during Hebbian and homeostatic plasticity events that modify the synaptic strength [3]. The onset of symptoms associated with ASD, ID, and ADHD coincides with the peak of cortical spinogenesis occurring between late childhood and early adolescence [4]. These data provide evidence for a neuroanatomical basis of these pathological states associated with alterations in dendritic spine maturation. Several experimental approaches such as in vivo high-resolution microscopy, electrophysiology and optogenetics have been used to study the activity-dependent remodeling of dendritic spines during synaptic plasticity processes in pyramidal neurons of the neocortex and hippocampus [5,6]. Recently, several large-scale correlation analyses of neuroimaging and genetic data emerged from the Enhancing NeuroImaging Genetics through Meta-Analysis (ENIGMA) consortium and have allowed the identification of singularities and shared neuroanatomical alterations in neurodevelopmental and neuropsychiatric disorders such as ASD, ID, schizophrenia (SCZ) and ADHD [7].

## 2. Attention Deficit-Hyperactivity Disorder (ADHD)

ADHD is a neurodevelopmental disorder with high prevalence (around 5%) among children worldwide characterized by hyperactivity, inattention and/or impulsivity affecting learning and sociability at school [8]. Several studies support a polygenic cause for ADHD and non-genetic factors involved in the etiology of this disorder. Environmental factors such as smoking and alcohol consumption during pregnancy, exposure to contaminants such as lead, diet deficiencies and low educational attainment have been correlated with ADHD [9,10,11]. In the 40–60% of children with ADHD, the disorder persists into adulthood [12]. Furthermore, there is co-occurrence with psychiatric disorders such as depression and anxiety [13,14].

Several neuroanatomical studies support the view that ADHD is a neurodevelopmental disorder. A significantly lower cortical thickness (most prominent in the prefrontal cortex, PFC) in children with ADHD as evaluated by brain MRI has been reported. The fitting of curves of thickness values obtained from sequential brain scanning in children diagnosed with ADHD and healthy controls show a delay of ≈3 years to attain the peak of cortical thickness during childhood, supporting a delay in cortical maturation in ADHD [15].

Meta-analysis of neuroimaging data obtained in children, adolescents and adults with ADHD show a statistically significant reduction (compared to control groups) in cortical thickness, cortical surface area and volume of subcortical areas such as the accumbens, amygdala, caudate, putamen and hippocampus, in children with ADHD. Interestingly, the only brain area significantly reduced in adolescents is the hippocampus [16]. Interestingly, analysis of brain MRI data from 145 groups of patients crossing six psychiatric disorders including ADHD, reveal a shared profile of differences in cortical thickness [17]. It is noteworthy that the cross-disorder correlation analysis shows a positive value for differences in cortical thickness and gene expression between ADHD and major depressive disorder (MDD). Recently, a positive correlation between neuroanatomical parameters such as gray matter volume in neocortex and basal ganglia and the individual scores in N-back task (to evaluate working memory) in ADHD adolescents has been reported [18].

Extensive evidence supporting a polygenic origin of ADHD with additional environmental risk factors has been reported [11]. A recent genome-wide analysis has uncovered 12 significant risk loci in ADHD. The biological activities in these loci include neuronal differentiation (*FOXP2*), glutamate receptor trafficking and anchorage to postsynaptic density (*SORCS3*), regulation of dopamine transporter trafficking (*DUSP6*) and axon guidance (*SEMA6D*) [19]. The same research group presented an updated genome-wide association study (GWAS) increasing the number of ADHD cases included in the meta-analysis, identifying 27 risk loci including 6 of those previously reported by the same research group [20]. Among these genes, mainly expressed in the frontal cortex (excitatory and inhibitory neurons), were found *FOXP1* and *FOXP2* (encoding transcription factors) previously involved in cannabis use disorder and intellectual disability [21,22]. By another part, a familial ADHD phenotype caused by a missense mutation in the *CDH2* gene, coding for the adhesion protein N-cadherin involved in synaptogenesis, has been recently described [23]. Knock-in transgenic mice carrying this mutation exhibit behavioral and cognitive phenotypes associated with ADHD. In addition, CA1 pyramidal neurons contained in brain slices from these animals display a lower frequency of spontaneous miniature excitatory postsynaptic currents (mEPSC) without changes in the amplitude of currents, suggesting changes at the presynaptic level. Furthermore, CDH2-KO mice exhibit lower levels of tyrosine hydroxylase (TH) transcripts (qPCR) with a significant reduction in TH-positive neurons (immunohistochemical analysis). These results, considered together with the reduced dopamine concentration in PFC of mutant mice, support the effect of N-cadherin mutation on activity levels of dopaminergic pathways and its relevance for ADHD-associated neurophysiology.

ADHD is associated with a dysfunction of dopamine and noradrenergic systems involving cortical areas such as the PFC (dorsolateral and ventromedial), cingulate cortex and basal ganglia (nucleus accumbens, caudate nucleus and putamen) affecting neural networks driving executive control, motor function, reward processing and decision-making [24]. The pharmacological therapies approved by the American Food and Drugs Administration (FDA) for the treatment of ADHD include stimulants such as amphetamines and methylphenidate (MPH), and nonstimulants as atomoxetine (ATX). Amphetamines and MPH increase synaptic levels of dopamine and norepinephrine by inhibiting DAT and NET symporters, whereas ATX selectively inhibits the NET [25]. The analysis of the dopamine levels in different brain areas by positron emission tomography (PET) imaging based on the competition between endogenous dopamine and a synthetic radioligand of the D2/D3 dopamine receptor, show that MPH administration in ADHD patients decreases the radioactivity-associated signal in hippocampus to a lower magnitude compared to healthy controls. These data suggest a lower MPH-induced release of dopamine in the hippocampus of ADHD patients compared to healthy controls, suggesting an underactivation of dopamine pathways in this brain area [26]. Considering drug efficacy and tolerability, the evidence that emerged from a network meta-analysis supports as first-choice medication, the use of methylphenidate in children and adolescents, and amphetamines in adults. Clonidine and guanfacine, activators of α2-adrenergic receptors, that are also used in ADHD pharmacotherapy, exhibit lower clinical efficacy in children (see Table 1) [27]. Among non-genetic factors involved in the development of ADHD are maternal smoking during pregnancy and exposure to environmental contaminants as lead and pesticides [9,10]. Several animal models of ADHD using zebrafish, mice and rats have been developed allowing the study of cellular and molecular mechanisms underlying the action and efficacy of newly generated drugs for the treatment of the disorder [28]. Rodent models of ADHD can be grouped into three types: genetic models such as *SNAP-25* KO mice, *DAT* KO mice, *LPHN-3* KO mice and rats, and those induced by exposure to environmental chemicals such as lead and pesticides, and by prenatal exposure to nicotine or alcohol [29]. The prenatal nicotine exposure (PNE) murine model exhibits several symptoms described for ADHD patients, such as increased locomotor activity, low performance in cognitive tests, inattention, and transmissibility until the third generation. In addition, PNE mice exhibit a reduction in dopamine turnover in the frontal cortex and striatum and a significant volume reduction in the cingulate cortex [30,31,32,33,34].

## 3. Long-Term Potentiation and ADHD

Excitatory transmission at glutamatergic synapses involves the activation of two types of glutamate-gated ionotropic receptors that produce excitatory post synaptic currents (EPSCs). Glutamate released into the synaptic space activates the α-amino-3-hydroxy-5-methyl-4-isoxazolepropionic acid receptors (AMPARs) present at the postsynaptic membrane, allowing the influx of sodium and calcium ions which shift the resting potential to a more positive value. This change in membrane potential relieves the magnesium block of the highly Ca^2+^-permeable *N*-methyl-D-aspartate receptors (NMDAR), inducing postsynaptic depolarization [35]. High frequency stimuli trigger long-term potentiation (LTP), that involves the insertion of AMPARs into the postsynaptic membrane of dendritic spines which increases the synaptic efficacy [36]. AMPARs in the postsynaptic membrane exist as homo or heterotetramers composed of two different subunit types: GluA1, GluA2 or GluA3. In the hippocampus, the AMPARs are present in the post-synaptic density of dendritic spines, mainly as GluA1/GluA2 and GluA2/GluA3 heteromers and a small fraction of as GluA1/GluA1 homomers. GluA2-containing AMPA receptors show linear I-V relationships and are Ca^2+^-impermeable. By contrast, receptors lacking GluA2 subunit are Ca^2+^-permeable and exhibit inward rectifying currents due to the block by intracellular polyamines at positive potentials [37]. During LTP, the process of translocation and fusion of AMPAR-containing vesicles involves the phosphorylation of residues Ser831, Ser845 and Ser818 at the GluA1 subunits of receptors. The shift in membrane potential of postsynaptic neurons relieves the magnesium-dependent block of NMDARs and allows calcium influx, activating CaMKII and PKC, and inducing the phosphorylation of residues Ser831 and Ser818 of AMPARs, respectively [36]. The phosphorylation of Ser845 residues by PKA has been implicated in trafficking of receptor-containing vesicles during LTP and is increased by the activation of an adrenergic receptor-dependent pathway [38,39]. After translocation and fusion of AMPAR-containing vesicles to the extrasynaptic membrane, they are mobilized by lateral diffusion to the post-synaptic density and anchored by the scaffold protein PSD-95 and transmembrane AMPAR regulatory proteins (TARPs) [40]. The current model proposed for high frequency stimulus (HFS)-induced hippocampal LTP involves an early phase associated with the lateral diffusion of AMPA receptors to the post-synaptic density and a late phase involving the exocytosis of receptor-containing vesicles to the extrasynaptic membrane domain [41].

Analyzing the effect of MPH on learning and hippocampal LTP at the molecular level, it has been reported that a single oral administration of MPH enhances hippocampal LTP and improves visuo-spatial learning (evaluated in the Morris water maze test) in rats. At the molecular level, this effect was associated with increased levels of GluA1-containing AMPARs at the cell surface, and higher levels of phosphorylated receptors at S845 and S831 residues. In addition, whole-cell recordings in CA1 neurons show higher EPSC amplitudes and enhanced short-term plasticity associated with a lower decay time of currents elicited by 20 Hz stimulation protocols [42]. Considering these results supported by pharmacological studies, we proposed a model for the action of MPH on hippocampal LTP in which the drug-induced increase in noradrenaline levels at the synaptic space activates dopaminergic terminals, inducing the activation of D1/D5 receptors on the postsynaptic CA1 glutamatergic neurons and the downstream activation of protein kinase A (PKA) in the dendritic spine. This cascade enhances the fraction of PKA-induced phosphorylation of GluA1-containing AMPARs at S845, increasing the surface levels of receptors and synaptic efficacy of the CA3-CA1 circuit. In addition to the HFS-induced increase in the number of AMPARs (LTP_N_) at the postsynaptic membrane, a novel mechanism involving a PKA-dependent increase in unitary conductance of AMPARs (LTPγ) has been recently reported [43].

Among symptoms associated with ADHD are the working memory impairments in children [44]. However, little is known about the neurophysiological basis of this memory deficit. Recently, using the PNE murine model of ADHD it has been found that the defect in working memory is associated with a significant reduction in LTP at the CA3-CA1 hippocampal synapse [31]. In addition, whole-cell patch clamp recordings of AMPA- and NMDA-dependent EPSCs from CA1 pyramidal neurons have shown a significant reduction in AMPA current amplitude in PNE animals, compared to control CA1 neurons. Biophysical properties of AMPAR-dependent EPSCs in PNE neurons suggests a change in the subunit composition of synaptic AMPARs to an enhanced fraction of GluA2-containing receptors, explaining the lower rectification index of AMPAR-dependent EPSC compared to control neurons. The electrophysiological data are supported by Western blot analysis showing that surface GluA1 subunit levels are decreased in CA1 neurons contained in slices from PNE mice, associated with lower levels of phosphorylated GluA1 subunits in residues S845 and S831 after LTP induction. These changes were not observed in PNE mice after a single administration of MPH [30].

## 4. Dendritic Spines and ADHD

Several human neurodevelopmental and neuropsychiatric disorders such as ASD, SCZ and ADHD are correlated with abnormal development and maturation of dendritic spines in the neocortex including changes in the pruning phase occurring during late childhood and early adolescence [3]. Dendritic spines are dynamic structures mainly associated with excitatory synapses exhibiting changes in density, morphology and functionality during fetal and postnatal development and activity-dependent remodeling events [45]. The time course of spinogenesis during fetal and postnatal development in humans and rodents (PFC and hippocampus) can be described by three phases: spine density increases and maturation until early childhood, spine density decreases during adolescence (associated with activity-dependent spine pruning or elimination) and stabilization in adulthood [3]. Dendritic spines have been classified into three to five different types as long-thin, thin, filopodia, stubby and mushroom according to the morphological criteria, head diameter and neck length, and associated with different states of the spine maturation process [45]. Filopodia type spines are highly dynamic structures present mainly during early postnatal development and almost absent in the adult brain, exhibiting lifetimes from minutes to hours. The mushroom type displays mature and larger structures and is the most abundant spine type in the adult brain with lifetimes as long as one year. From in vivo two-photon imaging in pyramidal cells of the visual cortex, lifetime values for dendritic spines between hours, days and several months have been estimated [46,47,48]. Although the classification of spines in different subtypes (between two or five types) considering morphological parameters is generally accepted, a recent analysis emerged from high-resolution EM images and 3D spine reconstruction, showing the unimodal and continuous distribution of spine parameters such as head volume and neck diameter suggesting the lack of defined morphological subtypes of spines [49].

Single spine stimulation with uncaged glutamate in hippocampal slices cultured and transfected with enhanced green fluorescent protein (EGFP) and whole-cell patch clamp recordings have shown that spine volume and AMPA currents are directly correlated. Small new spines contain low AMPA-dependent EPSCs and bigger persistent spines exhibit higher AMPA-EPSCs. In this cell model, NMDA-dependent EPSCs are not significantly different in spines of different size and lifetime [50]. A recent molecular analysis of spine dynamics using high-resolution fluorescence microscopy, electron microscopy and quantitative biochemistry in cultured hippocampal neurons has shown that the proteome of mushroom spines are better associated with synaptic strength compared to stubby spines [51].

The intracellular cascade involved in the structural and functional remodeling of dendritic spines during processes of synaptic plasticity has been studied mainly by investigating the effect of knock-out or block of specific proteins during the induction of LTP or long-term depression (LTD) [52]. In basal conditions, the proteins involved in synaptic transmission are clustered in postsynaptic densities, whereby scaffold proteins such as PSD95 allow the anchorage of membrane receptors, auxiliary subunits, kinases, and phosphatases [53]. The high frequency activity-triggered modification of ionotropic glutamate receptors’ distribution by exocytosis and lateral diffusion triggered by calcium increase induces the polymerization of F-actin filaments, inducing the spine head enlargement. LTP-induced spine enlargement (also known as structural LTP, sLTP) involves NMDAR-mediated calcium increase, CaMKII activation, and downstream stimulation of small GTPases (such as RhoA, Ras, Cdc42 and Rac1), leading to the fast remodeling of F-actin [54]. The F-actin dynamics based on the polymerization/depolymerization equilibrium is controlled by the interaction with cofilin-1 during the activity-induced spine remodeling [55].

## 5. Dendritic Remodeling and Learning

Studies focused on the maturation, homeostasis, and regulation of dendritic spines in the mammalian brain have been performed mainly in the neocortex, due to the limited resolution of microscopic tools for imaging of deep brain structures such as the hippocampus. Early studies were focused on mapping the sensitivity to uncaged glutamate of CA1 pyramidal neurons contained in slices and suggest that mushroom-type spine structures are correlated to a high density of AMPAR-dependent EPSCs—by whole-cell recording—reflecting functionally mature spines [56]. Later, the same group reported that repetitive uncaging of glutamate or repetitive stimulation of Schaffer collateral fibers induces a transient enhancement of spine heads supporting a structural basis for LTP [57].

The recent development of improved tools of two photon (2p)-microscopy has allowed the study of the spine dynamics in vivo in long-term imaging in the hippocampus, supporting a higher turnover compared to the neocortex. The mathematical modeling of spine kinetics in CA1 pyramidal neurons supports a single spine population with a medium lifetime of ~10 days in the hippocampus, in contrast to the long (up to several months) and short (~4 days) lifetime populations described in somatosensory and motor neocortices [6]. In addition, the opposite effect of the NMDAR blocker MK801 on spine density in hippocampal (increase in spine elimination) and cortical neurons (decrease in spine loss) suggest different molecular mechanisms are involved in spine dynamics in these two brain areas [6]. Comparative analyses of molecular pathways involved in hippocampal and cortical LTP also provides evidence for the existence of different mechanisms underlying the synaptic plasticity in these areas. LTP induced in the visual cortex is abolished by PKA inhibitors, in contrast to the hippocampal LTP that is sensitive to these inhibitors only during postnatal development [58].

The intracellular pathways implied in the sLTP involve NMDARs opening, high increases in cytosolic Ca^2+^, Ca^2+^/calmodulin-dependent protein kinase II (CaMKII) activation, Ras/RhoA/Rac1/Cdc42 activation, phosphorylation of cofilin, and polymerization of actin filaments that determine the spine head enlargement. Analysis of single spine dynamics in CA1 neurons show that the repetitive glutamate uncaging induces an increase in spine head volume, which exhibits an initial transient component followed by a long-lasting component. Both phases are inhibited by AP5 (a selective NMDAR antagonist) and only the sustained phase by KN62 (a selective CaMKII inhibitor), but fully abolished by latrunculin A (an inhibitor of actin polymerization), supporting the role of NMDAR-dependent calcium influx, the late activation of CaMKII, and the downstream remodeling of the F-actin network in spine remodeling [57]. The activation of CaMKII during LTP has been directly demonstrated by FRET measurements at the single spine level in hippocampal neurons transfected with a CaMKIIα labeled with donor and acceptor fluorophores at the N- and C-terminals of the enzyme [59]. The LTP induced by pairing two-photon glutamate uncaging and postsynaptic depolarization (by whole-cell patch clamp) of a pyramidal neuron can trigger the transient (and localized) activation of CaMKII and an increase in the volume of stimulated spines. This activation is dependent on L-type voltage-sensitive calcium channels and is abolished by BAPTA dialysis in the whole-cell configuration, suggesting that calcium nanodomains near calcium channels are sufficient to activate CaMKII in spines and dendrites. On the other hand, the molecular events associated with low-frequency stimulation-induced LTD involve NMDARs opening, a low increase in cytosolic calcium, activation of calcineurin (a Ca^2+^/calmodulin-dependent phosphatase), dephosphorylation of cofilin and depolymerization of actin filaments that induces the spine shrinkage (see Figure 1). Recently, a mechanism of local autophagy-mediated degradation of synaptic proteins during chemically induced LTD in cultured hippocampal neurons has been also suggested [60].

The canonical view of excitatory synaptic transmission involves calcium-dependent neurotransmitter release from the presynaptic terminal that induces depolarization of the postsynaptic neuron that triggers the calcium-dependent spine enlargement. Interestingly, following SNARE assembly associated vesicular fusion by fluorescence resonance energy transfer (FRET) in cultured hippocampal slices, it has been reported that 2P glutamate uncaging-induced spine enlargement enhances presynaptic exocytosis [61].

Several studies demonstrate a significant spine enlargement and formation of new spines in cortical neurons induced by motor skill learning in mice [62,63]. Fu and coworkers, analyzing the spine dynamics in layer 5 pyramidal neurons of the motor cortex by in vivo two photon-microcopy have shown that acquisition of motor skills is associated with the generation of new spines organized in clusters [64]. This training-induced clustered spine fraction (15.8%) exhibits a significantly larger lifetime and increases in head diameter compared to those non-clustered new spines. The direct relationship between spine remodeling and learning-associated synaptic plasticity has been confirmed by in vivo experiments investigating the effect of optogenetically induced shrinkage of task-potentiated spines on motor memory. Hayashi-Takagi and coworkers reported that artificially induced spine shrinkage by selective photoactivation of a Rac1 variant in active synapses is able to erase the acquired motor learning in a rotarod task [65]. Conversely, it has been reported that the increase in stability of newly formed spines in motor cortex enhances motor learning, after analyzing the spine dynamics and learning behavior in the *PirB* (paired immunoglobulin receptor B) knock-out mice [66]. This effect is explained by the requirement of PirB receptor protein—expressed in hippocampal and cortical pyramidal neurons—for NMDA-dependent spine shrinkage in cortical neurons. In addition, the motor learning-induced spine remodeling found in the motor cortex has also been documented in emotional learning and memory processes. Recently, Xu et al. reported that during fear conditioning there is spine elimination in the apical dendrites of layer 5 pyramidal neurons of the motor cortex. By contrast, the formation of new spines is observed during the fear extinction period [67].

Several studies report mechanisms of spine remodeling induced by growth factors, hormones and drugs [68]. Analyzing the uncaged glutamate-induced sLTP in cultured hippocampal slices using a FRET assay and 2p-microscopy to evaluate the role of brain-derived neurotrophic factor (BDNF) signaling, Harward and coworkers reported that glutamate uncaging on single spines induced the activation of tropomyosin receptor kinase B (TrkB), the membrane receptor for BDNF. In addition, spine enlargement was significantly blocked by the presence of an anti-TrkB neutralizing antibody during the sLTP induction, suggesting the requirement of BDNF for the glutamate-induced spine remodeling [69]. The inhibitory effect of specific blockers of NMDAR (AP5) and CaMKII (CN21) on TrkB activation and the release of BDNF supports the role of a TrkB-dependent intracellular cascade involving NMDAR-dependent calcium influx and activation of CaMKII. Recent evidence also involves the TrkB-dependent activation of PKC and downstream Rac1 activation, triggering the fast remodeling of F-actin during the sLTP [70]. Furthermore, cycloheximide can abolish the uncaged glutamate-induced spine enlargement, suggesting that protein synthesis is required for fast spine remodeling [71].

The effect of ketamine as an antidepressant has been proposed to be caused by disinhibition of glutamate synapses by block of the NMDAR in inhibitory GABAergic interneurons. The restored glutamatergic transmission would activate the autocrine BDNF signaling in the postsynaptic neurons by the downstream activation of the mammalian target of rapamycin (mTOR), inducing the local translation of synaptic proteins and spine growth [72]. Additional mechanisms for the antidepressant effect of ketamine such as the NMDAR blockade-dependent increase in BDNF translation induced by the eukaryotic elongation factor 2 (eEF2) and the direct binding of ketamine to TrkB receptor have been recently reported [73].

The psychedelic drug psilocybin has been also proposed as pharmacotherapy for depression. It increases spine density and spine size in frontal cortex pyramidal neurons, as evaluated in vivo using 2p-microscopy in Thy1-YFP mice. This effect persists for as long as one month [74]. Recently, the release and autocrine signaling of insulin-like growth factors 1 and 2 (IGF1 and IGF2) secreted by hippocampal pyramidal neurons contained in organotypic slices has been reported. The conditional gene ablation of IGF1 decreases the glutamate uncaging-induced spine enlargement and HFS-induced LTP measured in single CA1 neurons without affecting the spine structural plasticity and LTP of CA3 neurons. Instead, the conditional ablation of IGF2 impairs dendritic spine enlargement and LTP in single CA3 neurons without affecting the structural and functional LTP of CA1 neurons. These results provide evidence for a significant role of IGF1 and IGF2 in hippocampal synaptic plasticity. Future studies are required to confirm the relevance of IGF1/IGF2 signaling pathways in synaptic plasticity processes in vivo [75].

Several neuroanatomic studies at the cellular level suggest a common neurodevelopmental basis for neuropsychiatric disorders such as ADHD, ASD and schizophrenia, including increased or decreased spine density due to abnormal spine pruning during development [76]. Post-mortem analysis of brain slices from patients with schizophrenia have shown low spine density in dorsolateral PFC neurons probably associated with excessive spine pruning during development [77,78]. Recently, it has been reported that the overexpression of the human variant of C4A (complement component 4) in mice reduces spine density and enhances microglial-dependent synapse engulfment and pruning in the medial PFC. In addition, the C4A-overexpressing mice exhibit altered social behavior, an anxiety-like phenotype and impairment in spatial working memory, evaluated in the three-chambers test, open-field-test and novel Y-maze, respectively [79]. These results support the association of complement system-associated genes and risk of schizophrenia [80]. On the other hand, ASD has been associated with a defective spine pruning in layer V pyramidal neurons. Golgi staining of samples of postmortem human temporal lobes from ASD patients shows a significantly lower rate of decrease in spine density (due to spine pruning) occurring between 2 and 18 years as compared to healthy people [81]. In addition, the analysis of spine density in the haploinsufficient *Tsc2*^+/−^ mice defective in mTOR-dependent neuronal autophagy has revealed a deficit in synaptic pruning similar to that found in ASD patients [82].

Until now there are no reports focused on the analysis of spinogenesis during the childhood and adolescence stages in ADHD patients. The analysis of dendritic spine density and morphology in CA1 pyramidal neurons contained in Golgi-stained brain sections of PNE mice has demonstrated that hippocampal CA1 neurons from these animals exhibit lower spine density compared to control animals. In addition, there is a significant increase in the fraction of thin spines (immature) and a decreased fraction of mushroom ones (mature) compared to the percentage of types of spines in control mice [30]. These results point to a delay in the development of the hippocampus in PNE mice, as has been described for ADHD in humans [15]. It is noteworthy that CA1 neurons from PNE mice treated with a single dose of MPH exhibit no significant change in spine density compared to neurons of untreated PNE animals. However, MPH induces a significant decrease in the fraction of thin-type immature spines and a significant increase in the fraction of mushroom-type mature spines, suggesting a fast stimulatory effect on maturation status of dendritic spines in hippocampal neurons. The effect of chronic administration of MPH on dendritic spine density has also been documented in nucleus accumbens neurons [83]. Using a stress-induced model of ADHD, the MPH-induced recovery of dendritic spine density and length and branching complexity of dendrites in neurons of dorsal anterior cingulate cortex has been also reported [84].

## 6. Discussion

The recent efforts to elucidate the neurobiological basis of the neuropsychiatric and neurodevelopmental disorders such as ADHD have been focused on the meta-analysis of worldwide data of brain imaging from patients and analyzing the effects of mutations and the identification of risk loci associated with the disorders. On the other hand, basic research in animal models of these disorders serves to improve our understanding of the mechanisms underlying these pathological states, and their use in preclinical studies for testing drug efficacy continues to be fruitful. However, the remarkable differences of neurocytological organization of the neocortex and neuronal cytoarchitecture between mice and humans, restricts the studies of neurodevelopmental and neuropsychiatric disorders in murine models in terms of effective translation [85,86]. Comparative single-cell transcriptomic analysis of human and mouse cortical neurons show that serotonin receptor subunits (HTR3A and HTR3B) are among the most-divergent genes in terms of expression [87]. Furthermore, single-cell transcriptomic analyses in macaque and human brain areas during prenatal and postnatal development show a higher and more significant divergence in transcriptional profiles at prenatal and adult stages [88].The recent development of human organoids derived from stem cells opens a new alternative for in vitro modeling of diseases, cell therapy and drug development [89]. Using human telencephalic organoids generated from stem cell-derived single neural rosettes, Wang and coworkers have analyzed brain organoids carrying a hemizygous deletion of an autism- and intellectual disability-associated gene *SHANK3*. Patch-clamp recordings show that *SHANK3*-deficient organoids exhibit neuronal hyperexcitability and whole RNA-seq analysis supports that protocadherin and cadherin-dependent pathways are downregulated. The previous evidence pointing to a role of *PCDHA* (coding for protocadherin α) as a susceptibility gene for ASD [90], validates this organoid-based in vitro system for the research of neurodevelopmental diseases as well as for modeling and further drug development [91].

A relevant unsolved question in Neuroscience is to determine the neuroanatomical cues involved in the acquisition of higher cognitive functions during the evolution of mammals. Recently, the presence of an anterior–posterior gradient of retinoic acid in primate brain during development, enriched in PFC and involved in neocortex fetal development, has been reported. Interestingly, the gradient of retinoic acid is absent in the mouse brain [92]. Among the downstream targets of retinoic acid during neurodevelopment was identified the synaptic organizer cerebellin 2 (CLBN2), whose expression is also enriched in the human PFC. The *CLBN2* gene contains multiple cis regulation sites of repression mediated by Sox5 (a transcription factor), explaining the reduced spinogenesis in PFC neurons in rodents. The *CLBN2* gene in humans lacks Sox5 binding sites (in two gene enhancers) making the expression of this organizer insensitive to inhibition by Sox5, allowing a higher spine density in the human PFC. These findings would support a structural basis for higher cognitive functions in humans [93].

These results contribute to uncovering the neuroanatomical cues involved in the acquisition of higher cognitive functions in humans during primate evolution. In this context, recent studies focused on the development of the neocortex during the recent evolution of Hominidae have been reported. Comparative analysis of genomes of modern humans and Neanderthals identified a single amino acid substitution in the transketolase-like 1 (TLTK1) protein, an enzyme required for fatty acid synthesis. The variant found in modern humans determines the proliferation and growth of basal radial glial cells, progenitors localized in the subventricular zone, mainly promoting cortical neurogenesis in the frontal lobe [94]. Future molecular and cellular biology studies focused on elucidating the molecular events involved in human brain spinogenesis will be relevant to better understand the origin and progression of neurodevelopmental and neuropsychiatric disorders.

## 7. Conclusions and Futures Perspectives

It is well documented that some neuropsychiatric and neurodevelopmental disorders such as SCZ, ASD and ID exhibit abnormal spine density in the brain associated with abnormal spinogenesis or pruning during postnatal development. However, neuroanatomical studies focused on the development of dendritic spines in ADHD patients are missing. Morphological analysis of dendritic spines in hippocampal CA1 neurons in the PNE murine model of ADHD suggests an impairment in postnatal spine maturation. Further studies using in vivo spine imaging in CA1 pyramidal neurons of PNE mice are required to confirm the correlations between spine remodeling, impairment in hippocampal LTP and defective working memory.

Recent results that have emerged from GWAS using data from worldwide cases to identify risk loci for ADHD and brain MRI meta-analyses show that shared and unique neuroanatomical features will be relevant in the context of defining biomarkers for the disorder to improve early diagnosis, the identification of ADHD subtypes (inattentive, hyperactive and combined) as well as personalized drug therapy [8]. In addition, several studies support the influence of extrinsic factors such as diet, stress, and drug consumption during pregnancy on the development of ADHD [11]. Recently, the methylation state of the *DAT* gene has been proposed as an epigenetic marker of ADHD [95]. Future studies focused on global and gene-specific epigenetic modifications correlated to extrinsic factors during prenatal development and childhood as inducers of ADHD will be useful to identify the developmental mechanisms involved in these disorders.

## Figures and Tables

**Figure 1 biology-12-01241-f001:**
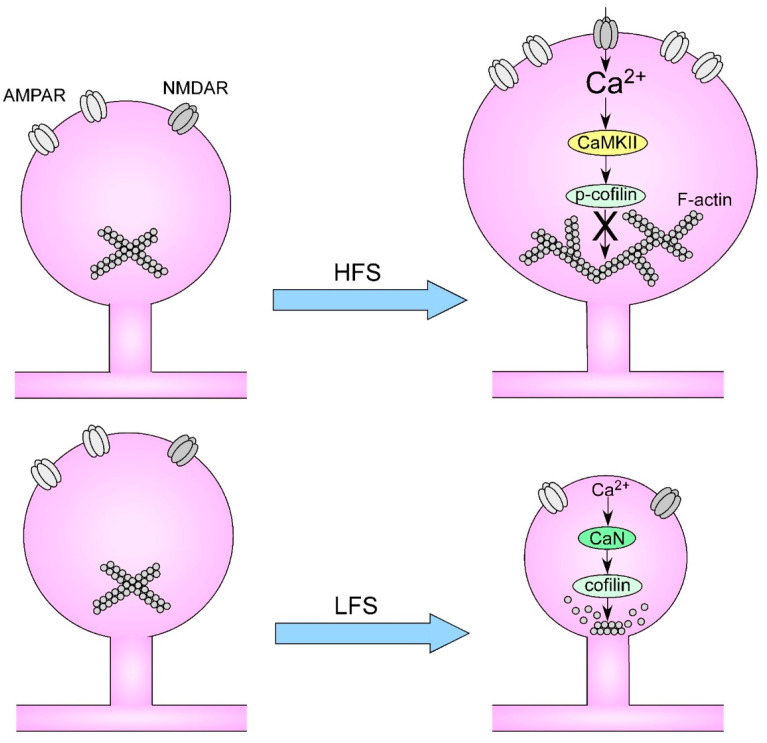
Schematic representation of intracellular pathways involved in the spine enlargement and shrinkage associated with high frequency stimulation (HFS) and low frequency stimulation (LFS) occurring after LTP and LTD, respectively. AMPAR: AMPA receptor; NMDAR: NMDA receptor; CaMKII: calcium/calmodulin-dependent protein kinase II; CaN: calcineurin; p-cofilin: phosphorylated cofilin. For details see the main text.

**Table 1 biology-12-01241-t001:** Medications currently used for the treatment of ADHD (Adapted from Cortese, 2020 [25]). DAT: Dopamine transporter, NET: Norepinephrine transporter, α2-AR: Adrenergic receptors; (+): activation, (−): inhibition.

Drug	Type	Molecular Target
Methylphenidate	Stimulant	DAT and NET (−)
Amphetamine	Stimulant	DAT and NET (−)
Atomoxetine	Nonstimulant	NET (−)
Clonidine	Nonstimulant	α2-AR (+)
Guanfacine	Nonstimulant	α2-AR (+)

## Data Availability

Not applicable.

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
