# Peer review of "Attention Deficit-Hyperactivity Disorder (ADHD): From Abnormal Behavior to Impairment in Synaptic Plasticity"

_biology, 2023, doi:10.3390/biology12091241_

Round 1
Reviewer 1 Report
The present review related to “Attention Deficit-Hyperactive Disorder (ADHD): from abnormal behavior to impairment in synaptic plasticity” is interesting. However, edits must be made before being considered for publication in Biology.
Edits and suggestions :
1. In the keywords, you could add Attention Deficit-Hyperactivity Disorder.
2. Line 14 : “Attention Deficit-Hyperactive Disorder (ADHD)is a neurological disorder”, it would be more precise to indicate neurodevelopmental disorder (DMS-5).
3. Lines 38-39 : “ADHD is considered as a neurodevelopmental disorder as Autism Spectrum Disorder (ASD) and schizophrenia (CZ)”. Schizophrenia is not yet considered as a neurodevelopmental disorder in the DSM-5, American Psychiatric Association. You can give an others examples of neurodevelopmental disorder.
4. Lines 59-60 “Among non-genetic factors involved in the causation of this disorder figures”. It would be more precise to indicate that genetic and environmental factors are implicated in ADHD etiology.
5. Lines 61-64 : “Further, it is known that around 60-70 % of patients exhibiting ADHD during childhood maintain the symptoms in adult age and can co-exists with psychiatric disorders as depression and anxiety [8].In the 40- 60% of children with ADHD the disorder persists into adulthood [11] and also exhibit co-occurrence with psychiatric disorders”. Please reformulate your sentence it's the same idea. Moreover, specify that the symptoms during childhood can co-exist with others neurodevelopmental disorders like ASD.
6. Line 348 : “for psychiatric disorders sch as ADHD, ASD and schizophrenia” Neuropsychiatric disorders are more precise instead of psychiatric disorders.
7. In future perspectives, it would be interesting to explain the future therapeutic perspectives.
8. Please, could you add a conclusion.
9. It would be interesting for the reader to put 1, 2 figures or tables in your review.
10. Your review should be enriched by more recent references (2022-2023).
Minor editing of English language required
Author Response
- In the keywords, you could add Attention Deficit-Hyperactivity Disorder.
It has added
- Line 14 : “Attention Deficit-Hyperactive Disorder (ADHD)is a neurological disorder”, it would be more precise to indicate neurodevelopmental disorder (DMS-5).
It was replaced
- Lines 38-39 : “ADHD is considered as a neurodevelopmental disorder as Autism Spectrum Disorder (ASD) and schizophrenia (CZ)”. Schizophrenia is not yet considered as a neurodevelopmental disorder in the DSM-5, American Psychiatric Association. You can give an others examples of neurodevelopmental disorder.
The intellectual Disability was included
- Lines 59-60 “Among non-genetic factors involved in the causation of this disorder figures”. It would be more precise to indicate that genetic and environmental factors are implicated in ADHD etiology.
Additional environmental factors were included, and the genetic causes are included in another paragraph with the recent results emerged from GWAS.
- Lines 61-64 : “Further, it is known that around 60-70 % of patients exhibiting ADHD during childhood maintain the symptoms in adult age and can co-exists with psychiatric disorders as depression and anxiety [8].In the 40- 60% of children with ADHD the disorder persists into adulthood [11] and also exhibit co-occurrence with psychiatric disorders”. Please reformulate your sentence it's the same idea. Moreover, specify that the symptoms during childhood can co-exist with others neurodevelopmental disorders like ASD.
The second sentence referring persistence into adulthood was deleted
- Line 348 : “for psychiatric disorders sch as ADHD, ASD and schizophrenia” Neuropsychiatric disorders are more precise instead of psychiatric disorders.
The adjective was rectified
- In future perspectives, it would be interesting to explain the future therapeutic perspectives.
That topic was commented in the new version of Conclusions and Future Perspectives
- Please, could you add a conclusion.
The section Conclusions and Future Perspectives was added
- It would be interesting for the reader to put 1, 2 figures or tables in your review.
One figure and one table were included in the manuscript
- Your review should be enriched by more recent references (2022-2023).
Two relevant reports published during the present year were included: the discovery of new risk loci associated to ADHD (Moliner et al, 2023) and the role of IGF1/IGF2-dependent signaling in spine structural plasticity and LTP (Tu et al., 2023)
Reviewer 2 Report
This review addresses the neurobiological substrate of ADHD, which may interest researchers both at clinical and preclinical levels. However, some important structural aspects need to be addressed; please see below:
General observations
-a thorough proofreading of this manuscript is required, because there are some recurrent grammatical and syntactical errors throughout the text. For example, “as” is used when “such as” would be preferable, and “associated to” is not a valid formulation, except for specific situations, the recommended formulation being “associated with”;
-CZ is not a customary abbreviation for schizophrenia; consider replacing it with SCZ or SCHZ;
-there are no objectives of this review mentioned in the body text, and no hints about the methodology used to collect and interpret data; also, the rationale for constructing this review is missing, maybe the Authors can insert some contradictions in the literature about the neurobiology of ADHD, or some yet unclear areas that may benefit from the current research;
-what are the strengths and limitations of this review, according to the Authors’ self-evaluation?
-a section about “Discussion” is needed, as well as a paragraph dedicated to the “Conclusions” of this review.
Specific observations
-please consider re-structuring the “Abstract” section, because the objectives are mentioned at the end of the “Abstract”; it is not clear what type of methodology you have used for collecting and reviewing the sources, and it is difficult to distinguish where the “Background” ends and where your conclusions start;
-lines 36 -38- it is recommended to rephrase that sentence, for clarity;
-line 69 - the formulation “at ages between 7-13 years old children” should be changed for clarity;
-line 70 - “between healthy controls and children with ADHD”;
-line 71- “By another part” is an improper formulation;
-line 109- “and whereas…” is incorrect;
-lines 135-136 “AMPARs…. exist…”;
-lines 135-147- are all these sentences extracted from the references (31) and (32)? If there are some verbatim reproduced sentences, then use quotation marks;
-the LTP abbreviation is not defined in the text (the definitions in the Abstract should be repeated in the text); please define it the first time you mention it; the same for LTD;
-line 178-181- please rephrase for clarity;
-line 267- “underlaying” is a typo;
-line 270- “…involved…. Involve…” is an unnecessary repetition, please rephrase; the same for lines 278-279;
-lines 281-285- any reference for these sentences?
Moderate English language editing is necessary.
Author Response
General observations
-a thorough proofreading of this manuscript is required, because there are some recurrent grammatical and syntactical errors throughout the text. For example, “as” is used when “such as” would be preferable, and “associated to” is not a valid formulation, except for specific situations, the recommended formulation being “associated with”;
Those errors were corrected
-CZ is not a customary abbreviation for schizophrenia; consider replacing it with SCZ or SCHZ;
That abbreviature was replaced
-there are no objectives of this review mentioned in the body text, and no hints about the methodology used to collect and interpret data; also, the rationale for constructing this review is missing, maybe the Authors can insert some contradictions in the literature about the neurobiology of ADHD, or some yet unclear areas that may benefit from the current research;
About the methodology used to collect and interpretate data, we searched in PubMed recent reports published in high-impact journals using as search terms such as “ADHD” “dendritic spine” “dynamics” “LTP” “learning” “neocortex” “hippocampus” “pharmacotherapy”.
Respecting to the open questions remaining in the field of ADHD and dendritic spine dynamics in such as the translational efficacy of animal model of ADHD and the physiological relevance of phenotypical classification of spines are included in Discussion.
-what are the strengths and limitations of this review, according to the Authors’ self-evaluation?
About the strength and limitations of this review, our work emerged as an attempt to review the recent reports focused on ADHD integrating multiple approaches such as genetics, pharmacotherapy, molecular neurobiology, preclinical models and evolutive biology. Considering the size limitations of the manuscript the full description of shared and exclusive cognitive deficits described for ADHD and other neurodevelopmental disorders were not included.
-a section about “Discussion” is needed, as well as a paragraph dedicated to the “Conclusions” of this review.
The section called Future Perspectives included in the previous version of manuscript was reformulated in Discussion and Conclusions and Future Perspectives
Specific observations
-please consider re-structuring the “Abstract” section, because the objectives are mentioned at the end of the “Abstract”; it is not clear what type of methodology you have used for collecting and reviewing the sources, and it is difficult to distinguish where the “Background” ends and where your conclusions start;
A Short Summary was added and the Abstract edited.
-lines 36 -38- it is recommended to rephrase that sentence, for clarity;
That sentence was rephased
-line 69 - the formulation “at ages between 7-13 years old children” should be changed for clarity;
That sentence was rephrased
-line 70 - “between healthy controls and children with ADHD”;
That sentence was rephrased
-line 71- “By another part” is an improper formulation;
It was deleted form the text
-line 109- “and whereas…” is incorrect;
It was corrected
-lines 135-136 “AMPARs…. exist…”;
It was corrected
-lines 135-147- are all these sentences extracted from the references (31) and (32)? If there are some verbatim reproduced sentences, then use quotation marks;
The description of the AMPA subunits composition and regulation during LTP was reformulated from the review corresponding to the reference 31. The reference 32 was erroneously included.
-the LTP abbreviation is not defined in the text (the definitions in the Abstract should be repeated in the text); please define it the first time you mention it; the same for LTD;
The abbreviations LTP and LTD were properly defined
-line 178-181- please rephrase for clarity;
That paragraph was rephrased
-line 267- “underlaying” is a typo;
That error was corrected
-line 270- “…involved…. Involve…” is an unnecessary repetition, please rephrase; the same for lines 278-279;
The first repetition was corrected, and the second paragraph was deleted by redundancy
-lines 281-285- any reference for these sentences?
That paragraph was associated to the reference 55
Round 2
Reviewer 2 Report
The manuscript substantially improved. Please consider the following additional observations:
Simple summary
Line 21- „dendritic spine” instead of just „spine” is a needed change for avoiding misunderstandings; also, consider changing „abord” with „approach”, because the first formulation is archaic;
Abstract
The structure of this section is still counterintuitive. The aim should be placed after one or two sentences of background, then a sentence about the methodology is recommended, followed by the results and conclusions of the Authors’ work. In its current form, the Abstract is confusing and the manuscript's contribution to the research in the field of ADHD is unclear.
Body text
Line 84- a patient-first formulation is recommended throughout the text, i.e., „children with ADHD” or „children diagnosed with ADHD” instead of „ADHD children”;
Line 214- please consider changing the expression „by another part”, as already suggested in the first round of reviewing, because it sounds awkward; possible replacements are „On the other hand”, „However”, „Otherwise”, „That being said”, etc.
Line 393- the same observation as before about replacing „spine” with „dendritic spines”;
Line 401- „patients with schizophrenia”;
The topic chosen by the Authors is very complex and, because there are no specified inclusion and exclusion criteria for their search in the electronic databases, important resources about this subject may have been missed, including papers exploring similar objectives (e.g., https://www.ncbi.nlm.nih.gov/pmc/articles/PMC2650365/, https://pubmed.ncbi.nlm.nih.gov/19621976/, https://pubmed.ncbi.nlm.nih.gov/28633952/). Maybe adding these methodological limitations in the „Discussion” section would help the readers understand why some data may be lacking from this review.
Minor editing problems have been identified.
Author Response
The manuscript substantially improved. Please consider the following additional observations:
Simple summary
Line 21- „dendritic spine” instead of just „spine” is a needed change for avoiding misunderstandings; also, consider changing „abord” with „approach”, because the first formulation is archaic;
It has been corrected
Abstract
The structure of this section is still counterintuitive. The aim should be placed after one or two sentences of background, then a sentence about the methodology is recommended, followed by the results and conclusions of the Authors’ work. In its current form, the Abstract is confusing and the manuscript's contribution to the research in the field of ADHD is unclear.
This work do not correspond to a systematic review or a meta-analysis, then the inclusion of Methodology and Results sections we considere not applicable.
The contribution of this review in the field of ADHD is to summarize the recent evidences emerged from several biomedical aproaches such as genetics, brain imaging, cellular biology and animal models, in order to evaluate the future strategies for the effective diagnosis and treatment of ADHD.
Line 84- a patient-first formulation is recommended throughout the text, i.e., „children with ADHD” or „children diagnosed with ADHD” instead of „ADHD children”;
It has been corrected
Line 214- please consider changing the expression „by another part”, as already suggested in the first round of reviewing, because it sounds awkward; possible replacements are „On the other hand”, „However”, „Otherwise”, „That being said”, etc.
That expression was replaced and the paragraph relocalized in the text.
Line 393- the same observation as before about replacing „spine” with „dendritic spines”;
It has been corrected
Line 401- „patients with schizophrenia”;
It has been corrected
The topic chosen by the Authors is very complex and, because there are no specified inclusion and exclusion criteria for their search in the electronic databases, important resources about this subject may have been missed, including papers exploring similar objectives (e.g., https://www.ncbi.nlm.nih.gov/pmc/articles/PMC2650365/, https://pubmed.ncbi.nlm.nih.gov/19621976/, https://pubmed.ncbi.nlm.nih.gov/28633952/). Maybe adding these methodological limitations in the „Discussion” section would help the readers understand why some data may be lacking from this review.
About the methodology used to collect and interpretate data, we searched in PubMed recent reports published in high-impact journals using as search terms such as “ADHD” “dendritic spine” “dynamics” “LTP” “learning” “neocortex” “hippocampus” “pharmacotherapy”.
Two additional references focused on the effects of methylphenidate on dendritic spines using animal models of ADHD were included.